# Functional Neurological Disorder–Old Problem New Perspective

**DOI:** 10.3390/ijerph20021099

**Published:** 2023-01-08

**Authors:** Radu-Stefan Perjoc, Eugenia Roza, Oana Aurelia Vladacenco, Daniel Mihai Teleanu, Roxana Neacsu, Raluca Ioana Teleanu

**Affiliations:** 1“Carol Davila” University of Medicine and Pharmacy, 020021 Bucharest, Romania; 2“Dr. Victor Gomoiu” Children’s Hospital, 022102 Bucharest, Romania; 3Emergency University Hospital, 050098 Bucharest, Romania

**Keywords:** functional neurological disorder, children, family

## Abstract

Functional neurological disorder (FND) is a common issue in the pediatric population. The concept and our understanding of functional neurological disorders have changed over the past years, and new etiologic models and treatment plans have been explored. Knowledge about FND in the pediatric population, however, is lacking. The aim of this review is to provide an update on pediatric functional neurological disorder. We conducted a literature search of PubMed and SCOPUS databases and reviewed a total of 85 articles to gain insight into the current understanding of FND etiology, diagnosis, treatment, and prognosis in children and adolescents. Functional and high resolution MRI revealed abnormal connectivity and structural changes in patients with functional symptoms. The diagnostic criteria no longer require the presence of a psychological factor and instead focus on a rule-in diagnosis. Treatment of FND includes a clear communication of the diagnosis and the support of a multidisciplinary team. Although FND typically has a poor prognosis, better outcomes appear to have been achieved in children and young adults. We conclude that pediatric functional neurological disorder is a prevalent pathology and that this patient population has additional specific needs compared to the adult population.

## 1. Introduction

Functional neurological disorder (FND) is a prevalent pathology in pediatric practice, characterized by neurological symptoms that are not caused by other classic neurological or medical conditions [1]. FND has an important socioeconomic impact on the adult population and is the second most common reason for neurological consults, following headache. The estimated incidence is 4–10/100,000, and is more common in adults [2,3,4,5,6,7,8,9]. In the pediatric population, the estimated incidence is 1.3/100,000, with both a higher incidence and prevalence in girls, the latter of which varies from 1 to 17 cases per 100,000 [6,10,11,12].

Children with FND present with a broad range of symptoms that can mimic those of neurological diseases, but the most common are seizures and motor-type symptoms (functional movement disorder) [2]. Motor-type FND, or functional movement disorder (FMD), is a subtype of FND that can present with positive symptoms (dystonia, jerks, tremor, etc.) or negative symptoms (weakness, paralysis). Although the incidence of FMD peaks in midlife, patients with functional dystonia and weakness seem to have a younger onset age than those with gait dysfunction [13]. Moreover, a clearly different distribution of FND by age is observed regarding gender [13]. A tendency for an early onset FND is noticeable in women, with a peak in late adolescence and a tapering down into adulthood [13]. In contrast, men have a similar FND symptom frequency across their lives [13]. Furthermore, the distribution of FND by gender shows that a greater amount of women experience FND symptoms across the lifespan compared to men [13]. Based on thorough history taking and clinical examination, the pediatrist can find incongruences between the patient’s complaints and their clinical signs. When suspicion of FND is raised, the medical practitioner should keep in mind that FND can coexist with an organic disease and that not all patients with FND have a history of stress factors or psychiatric comorbidities [14,15].

The aim of this article is to provide a review of FND and summarize the etiology, diagnostic approach, treatment options, and prognosis. There is little information about FND in the pediatric population in the medical literature, further emphasizing the need for higher quality studies to help better understand and provide care for these patients.

## 2. Materials and Methods

We present a narrative review on the current concepts of functional neurological disorders in children and young adults. We performed a literature search of PubMed and SCOPUS databases, and reviewed the current understanding of FND etiology, diagnosis, treatment, and prognosis in children and adolescents. Our search consisted of the following terms: conversion disorder, dissociative disorder, and functional neurological disorder. There were no restrictions by article type. The search was limited to articles published in English from 2000 up to September 2022. Our search resulted in 602 articles from which those containing the word “functional” unrelated to functional neurological disorder were removed. The remaining articles were screened by title and abstract, and 108 articles were chosen for a full-text review. For a comprehensive search, all relevant references from the selected articles were included regardless of the publication date. In total, 85 studies were included in the review. As in all narrative reviews, a selection bias has to be taken into account.

## 3. Results

### 3.1. Etiology

The etiology of FND is complex and no single causal mechanism has been found. The physical symptoms of patients with FND were previously believed to result from psychological distress, but recent studies show that only about one-third of these patients have a history of trauma [14]. Today, the etiology of FND is viewed as a complex combination of multiple predisposing factors, which vary from one patient to another. The predisposing factors most commonly associated with FND are: trauma/psychiatric symptoms; somatic symptoms; illness exposure; symptom monitoring; and neurobiological factors [16].

#### 3.1.1. Trauma

Trauma, stress, and psychiatric disorders are some of the most commonly recognized predisposing factors for the development of FND, but the research findings are inconsistent. A history of trauma, stress or psychiatric disorders has been found in only one-third of the adult population with functional symptoms [14,17]. A recent meta-analysis revealed a history of sexual abuse in 33% of patients with seizure-type FND [18], but no conclusive causal relationship could be drawn between these two factors. A study on the adult population found a history of childhood trauma of 70.3%, a much higher incidence than the previously reported 44% and a history of sexual abuse of 26.6% [19]. Antecedent stressors were found in 62% of children with FND by Kozlowska et al. [20] and in 81% of children by Ani et al. [12], which is much higher than in the general population. However, some studies suggest that patients with functional symptoms do not have a higher frequency of stressful life events than patients with epilepsy or the healthy population but instead, have greater self-reported distress caused by these events [2,21]. Patients with functional neurological disorder also have a higher activation of the stress system (hypothalamic-pituitary-adrenal axis) than the general population, in response to the same stress factor [22]. 

Although anxiety is reported as an important predisposing factor for FND, the evidence is inconsistent. Some studies have reported a higher prevalence of anxiety in these patients [23] while others have failed to find any correlation [16,24,25]. This may be partially explained by the fact that FND patients report a greater alexithymia [16].

The increased anxiety surrounding the COVID-19 pandemic was expected to aggravate FND symptoms, but no clear correlation was found. In a study conducted on patients with a history of FND, only a minority of patients showed a slight deterioration. Instead, there was an increase in the incidence of FND diagnosis [26,27]. One study showed a 90.1% increase in the pediatric population and a 50.9% increase in the adult population [27]. There was a rise in the number of patients presenting with FND during each phase of the pandemic and a relation with peak confirmed cases was found [26]. This correlation between number of cases and FND diagnosis might be explained by the increase in anxiety and depressive disorders caused by the lockdown and social distancing measures [26]. Furthermore, several cases of motor- and seizure-type FND were also reported after COVID-19 vaccination [28,29,30]. However, it should be noted that functional symptoms have also been reported after H1N1 and HPV vaccinations [31,32,33].

Although there is insufficient evidence to support trauma or psychiatric symptoms as the sole cause of FND, there is a higher prevalence of these factors and combined with other predisposing factors, they might increase the risk of developing functional symptoms.

#### 3.1.2. Illness Exposure 

The presence of physical trauma and somatic symptoms has been reported for a long time in patients with FND [34]. Children and young adults are more prone to physical injury, and the correlation between injury and functional symptoms might be hard to demonstrate. Nonetheless, 37% of patients report some form of physical injury prior to the development of FND [34].

Neurological disorders are an important risk factor for FND. A systematic review found a mean frequency of epilepsy of 22% in patients presenting with seizure-type FND and a mean frequency of seizure-type FND of 12% in patients suffering from epilepsy [35]. In almost all patients with a dual diagnosis, epilepsy occurred prior to FND [35]. 

A prior history of family illness also appears to be associated with FND. One study found a powerful relationship between reported poor health of the parents when the subject was an adolescent and development of symptoms later in life [36].

#### 3.1.3. Neurobiological Theory/Evidence

Given the wide heterogeneity of the predisposing factors found in patients with FND, multiple etiological models that integrate psychological, physical, and neurobiological factors have been published over the last few decades. The constructs that appear to be affected in FND include emotion processing, agency, attention, interoception, and predictive processing [37]. Evidence suggests that the presence of functional and/or structural abnormalities in the regions of the brain are involved in these constructs. Each construct might explain a different subtype of FND. For example, a decrease in self-agency (the feeling of being able to control external events through one’s own action) causes movements to be experienced as involuntary and might explain seizure-type FND [37]. Self-agency is caused by the matching of the feedforward signal (generated when a voluntary movement is produced by the primary motor cortex) and the feedback signal (generated by sensory experience when the movement occurs). One of the main sites for this matching is the right temporoparietal junction (TPJ). When a mismatch between these two signals occurs, the TPJ is activated. Studies have shown a right temporoparietal junction dysfunction in patients with motor-type FND [38,39], resulting in an abnormal functioning of the agency network in some patients.

Emotion processing involves the salience and limbic/paralimbic circuits. Functional studies have shown an increased connectivity between limbic and motor circuits in patients with functional neurological disorder, which might explain why negative emotions have been associated with worsening motor symptoms [40,41]. Patients with FND tend to have a heightened attention to their symptoms and an impaired ability to volitionally shift attention. The regions involved in attentional neurobiology are the perigenual anterior cingulate cortex [42], posterior parietal cortex, striatum, and thalamus [42]. Multiple functional studies have shown abnormal activity in these regions in patients with FND.

In addition to functional abnormalities, some studies have found structural abnormalities in patients with FND. For example, there is evidence of cortical atrophy in the right motor and premotor areas and the right and left cerebellum in patients with seizure-type FND [43]. Furthermore, two studies found decreased gray matter volumes in the thalamus and basal ganglia [44]. However, this decrease in volume was not found in the pediatric population [45], which suggests that decreased gray matter volumes in these regions might be related to illness duration. One study in the pediatric population demonstrated increases in the volumes of the left supplementary motor area (SMA), the right superior temporal gyrus (STG), and the dorsomedial prefrontal cortex (DMPFC). These areas are known to be involved in the perception of emotion and the modulation of motor responses [45].

### 3.2. Diagnosis

As the understanding of FND has improved, so has the diagnostic criteria. While the DSM-IV required the exclusion of other diseases and the presence of a psychological factor, the DSM-5 focuses on a positive diagnosis. The DSM-5 diagnostic criteria for FND include: (1) symptoms of altered voluntary motor or sensory function; (2) clinical findings incompatible with neurological or medical conditions; (3) symptoms that cannot be explained by another medical or psychiatric disorder; and (4) symptoms that cause significant distress or impairment in social, occupational, or other important areas of functioning [1].

Children with functional symptoms usually present for the first time to the general practitioner (GP) or pediatric emergency department (ED). These patients can have a wide spectrum of symptoms, such as motor function disturbances, sensory and visual dysfunctions (visual loss, diplopia, etc.), speech problems (dysarthria, mutism, foreign accent syndrome), and symptoms resembling epileptic seizures [46]. Motor symptoms are one of the most frequent types of presentations and can be divided into two broad categories: negative symptoms (weakness, paralysis) and positive symptoms (dystonia, tremor, jerks, etc.). Patients with sensory disturbances can have either numbness or positive sensory symptoms (tingling, itching, etc.). To better diagnose FND, medical professionals should educate themselves on patient history features and clinical signs that are more consistent with functional symptoms rather than an organic cause [46]. A recent review systemized positive signs for motor- and seizure-type FND, which can help clinicians make a faster and more accurate diagnosis [2] (e.g., distractibility, variability, monoplegic leg dragging, noneconomic posturing) with a specificity as high as 100%. For seizure-type FND, a report published by the ILAE describes the minimum requirements for diagnosis [47]. According to this report, signs favoring a FND diagnosis are fluctuating course, closed eyes, and ictal crying. The report also proposed diagnostic levels of certainty based on history, witnessed event, and EEG [47]. Video EEG, in combination with patient history and witnesses, serves as the gold standard for the differential diagnosis between seizure-type FNS and epileptic seizures. Braun et al. observed that compared to healthy controls, children with functional seizures maintain or even have an increase in beta rhythm post hyperventilation [48]. If, based on the history and positive signs, the general practitioner is confident in a FND diagnosis, a neurological consult may not be needed. It is important to remember that the diagnosis should not be based solely on the presence of a recent stress factor or psychological comorbidities given that these are not present in all patients [12,49].

Since early diagnosis is linked to a better prognosis [50], the medical provider should be educated in the delivery of the diagnosis to ensure patient and family acceptance regarding the disease. During the initial consultation, a thorough mental health evaluation should be performed to identify possible stress factors, history of trauma, or other psychiatric disorders. In any case, a complete evaluation should include a referral to a mental health specialist to help the patient cope with and treat the symptoms. Fear of missing an organic disease can lead to a prolonged interval from symptom debut to positive diagnosis and unnecessary treatment. For example, a study on 422 adult patients with a diagnosis of FND showed that 75% of the subjects that had only functional seizures (without epilepsy) had a positive history of antiseizure medication [51]. A study that assessed the accuracy rate of neurology residents of FND in the ED showed a correct diagnosis in 93–94% of patients [19,52]. Even though a false-positive result is rare in FND, medical professionals should not forget that FND can coexist with an organic disorder [53,54,55,56] and that the average rate of misdiagnosis of neurological disease as functional is 4% [52].

### 3.3. Communicating the Diagnosis to Patients and Families

Functional neurological disorder is a diagnosis that can elicit considerable stigma towards patients and their families. Moreover, the idea that these patients are “faking it” or that they are “not really sick” still exists in today’s society. Pediatricians face the challenge of having to explain the disease in a way that can be understood and accepted by both the patient and the family. Delivering the diagnosis is central to treating FND. For example, a study on 54 patients with functional seizures showed that 15 of these patients had a remission in symptoms after receiving their diagnosis [57]. Furthermore, a clear communication of the diagnosis has been found to reduce the use of healthcare services [58]. These factors indicate a clear need for medical practitioners to educate themselves on how to explain FND to their patients. During discussions with the patient and family, reviewing positive signs can help convince the patient that the correct diagnosis has been made. In these situations, it can be helpful to review videos of the clinical manifestations (taken by the family or the physician) together with the patient and his family so that they can see the positive signs that led to the diagnosis. Describing FND as a miscommunication between the brain and the body can also help the patient to better understand the disorder [2,59,60]. Here, the medical professional should use examples that are best suited to the patient’s level of knowledge and personal interests (e.g., explaining the disorder to a car enthusiast as an oil pressure sensor error rather than low oil levels). 

In our opinion, when possible, the diagnosis should be given to the patients in the presence of a mental health specialist. Several recommendations have been proposed to help the clinician deliver the diagnosis. Hall-Patch and colleagues developed a crib sheet to ensure that the clinician covers all the points in communicating the diagnosis. These include:

1. Recognizing symptoms as genuine;

2. Labeling the condition;

3. Providing a brief explanation (e.g., “a software rather than a hardware problem”; “the brain becomes overloaded and shuts down”); 

4. Addressing which treatments are effective and which are not;

5. Managing expectations (e.g., the patient can expect improvement; the disorder can be resolved) [59].

### 3.4. Treatment

Treating FND represents a challenge for both the healthcare professional and for the patients–from the delivery of the diagnosis to the selection of the best approach. Communicating the diagnosis in a clear and bias-free manner is the first step in treating these patients [2,3,60,61]. The next step is treatment through a multidisciplinary team that includes a neurologist, a psychiatrist, a psychologist, a rehabilitation medicine doctors, and a physiotherapist [60,61,62]. The usual treatment approach in the pediatric population includes Retraining and Control Therapy (ReACT), other cognitive behavioral therapies, and multidisciplinary rehabilitation [63,64]. Other trauma-specific interventions such as radical exposure tapping or Eye Movement Desensitization and Reprocessing (EMDR) can be used if required [65,66,67].

Cognitive behavioral therapy (CBT) is one of the most widely used interventions for treating functional symptoms and has shown a good response rate in the pediatric population [68,69]. A prospective study in children with seizure-type FND revealed a good efficacy of CBT, with 91% of the patients achieving partial or full remission after treatment [69]. However, this study did not specifically evaluate the efficacy of CBT as it was a part of a multidisciplinary approach. Nevertheless, CBT plays a crucial role in the treatment plan. CBT interventions should include education regarding the diagnosis, bottom-up regulation interventions, and family therapy. While psychotherapy is a cornerstone in treating FND, clinicians should be aware of coexisting psychiatric disorders, which should be treated according to the latest guidelines. 

Retraining and Control Therapy is a short-term outpatient intervention that uses cognitive behavioral principles to retrain catastrophic symptom expectations and low sense of control over symptoms. ReACT includes four steps: (1) a clear etiological description based on the Integrated Etiological Summary Model [16]; (2) an individually tailored patient plan to retrain physical symptoms which challenges catastrophic symptom expectations and teaches patients to engage in behaviors incompatible with PNES (similar to habit reversal), an evidence-based behavioral treatment for retraining tics; (3) a family plan to react to PNES in which they monitor the patient for safety but otherwise allow the patient to follow their plan to independently control the episodes; and (4) a plan to return to school and social activities [63]. Fobian et al. published a randomized controlled trial (RCT) that compared ReACT with supportive therapy in pediatric patients with seizure-type functional symptoms. In this study, ReACT showed significant reductions in symptoms and a rapid response when compared with supportive therapy. All children in the ReACT group achieved a complete resolution of symptoms at 7 days after therapy and 82% were symptom-free at 60 days post-treatment. The average number of sessions needed to reach zero seizures was 4.6 [63].

Physiotherapy plays a vital role in managing FND, especially in patients with motor symptoms. A randomized controlled trial on a group of 60 patients with motor FND revealed that physiotherapy shows high acceptability, good treatment effect, and a cost benefit. The follow-up period was 6 months and 72% of the patients in the intervention group showed a good outcome compared with 18% in the control group. A larger proportion of the control group (32%) felt that their symptoms worsened compared with 3% in the intervention group [70]. A systematic review of 29 articles showed a trend of positive outcome associated with physiotherapy in motor-type FND. In the pediatric population, 80% achieved a complete recovery and 16% a partial recovery. However, most of the studies included in the review had a small study population and occurred in an inpatient setting with patients receiving multiple interventions. Considering the apparent good results of physiotherapy in motor-type FND, a consensus was made to help choose a better physiotherapy intervention taking into consideration a biopsychosocial etiological framework [71].

Several studies have researched the efficacy of pharmacotherapy in FND but yielded no clear evidence of any beneficial purpose. A study on motor FND revealed that patients with FND and anxiety or depression might benefit from antidepressants [72]. However, the RCTs in patients with seizure-type FND failed to demonstrate a significant reduction in the number of episodes compared to placebo [73,74].

Functional studies have revealed evidence of abnormal connectivity in the central nervous system in patients with FND, which might be a consequence of poor neuroplasticity. Treatment options that aim to modulate this phenomenon are known to be effective in treating major depressive disorder and other trauma-related disorders. Several studies have evaluated the efficacy of transcranial magnetic stimulation (TMS) in adult patients with motor-type FND and found a good response to treatment, but others found only a small reduction in symptom severity. Although TMS appears to be of some benefit in treating motor-type FND, most of the studies on TMS have several limitations: the protocols used show wide heterogeneity; most studies had no placebo control group; and TMS was usually a part of a multidisciplinary approach.

Finally, physical activity (PA) is known to reduce depression and anxiety and increase overall quality of life [75,76,77,78] yet the data on the levels of PA in children and adolescents with FND are scarce.

### 3.5. Prognosis

Functional neurological disorder appears to have a generally unfavorable prognosis. A systematic review evaluated the outcomes of patients with motor FND and revealed that 39% (10–90%) of patient symptoms were the same or worse at follow-up evaluations [79]. Cohorts of patients with mixed functional neurological symptoms revealed that in 54% to 67% of patients, symptoms worsened or were unchanged at follow-up. However, many factors can influence the outcome of these patients. Pediatric studies reveal that a younger age is associated with a better prognosis. Remission rates in the pediatric population fall between 43% and 81%, and improvement of symptoms was found in 71–100% of patients [80,81]. In addition to age, symptom type appears to play an important role in the outcomes of these patients. One study found that children with motor- and seizure-type symptoms achieve better outcomes than those with sensory symptoms [81]. Duration of symptoms was also described as a predictive factor in a number of studies, with short duration being associated with a positive outcome [79,82]. A study on children with seizure-type FND found that a longer duration of the symptoms (over a year) is correlated with a six-time likelihood of failure to achieve full remission [82]. A higher education level/IQ is described in some studies as a positive prognosis factor [83], but the findings are inconclusive [84]. Although children with FND have a better prognosis than adults, there is still an important socioeconomic impact. One study on a cohort of patients with mean symptom duration of 7 weeks revealed an average of 22.4 days of school absence [19].

### 3.6. Difficulties in the Diagnosis and Treatment of FND

Although the diagnosis of FND is usually straightforward, the fear of missing an “organic” disease can make managing these patients difficult. This concern can lead to overuse of medical resources and inappropriate treatment. Over-investigation can cause considerable anxiety to patients and their families about a possibly serious disease. It can also lower confidence in the medical professional and thereby lead to a low compliance to further treatment. Inappropriate use of medical tests can lead to false-positive results and thus, a wrong diagnosis and treatment [85]. As previously mentioned, clear communication about the diagnosis is the most important step in the treatment of FND. In the pediatric population, this can be especially difficult as the child and both parents must accept the diagnosis and treatment plan.

## 4. Discussion

While the number of studies regarding the etiology and the management of functional neurological disorders has grown over the last several years, there is still a lack of large randomized controlled trials, particularly in the pediatric population. This narrative review provides an overall perspective on the latest findings in the field of functional neurological disorders. In the last decades, many studies have aimed to understand the pathophysiology behind FND. However, the mechanisms leading to functional neurological disorders are still not clearly understood nor is there a clear trigger of symptoms. Instead, etiological models that integrate neurobiological, physical, and psychological factors have been proposed. Owing to the structural changes identified in patients with functional symptoms, the term “non-organic” has become outdated and clinicians should avoid using it. These structural changes can be a sign of poor neuronal plasticity and treatment methods aimed at modulating plasticity, such as transcranial magnetic stimulation, might have a role to play in the management of patients with chronic FND. A better understanding of the pathophysiology and predisposing factors is needed to develop better treatment plans. 

The diagnosis of functional neurological disorder is usually straightforward and in most cases, no additional investigations are required. However, there is a 4% rate of a false-positive diagnosis and a thorough evaluation during the follow-up period is necessary to identify the patients who may require further investigations. Additional investigations may also help reassure parents of children with FND and raise acceptability as well as treatment compliance. 

Treatment of the disorder can include a wide spectrum of interventions. A multidisciplinary approach based on cognitive behavioral therapy with additional interventions is typically used. When devising a treatment plan, clinicians should take into consideration the duration of symptoms, psychiatric comorbidities, socioeconomic status, and accessibility to various treatment options. There are, however, no guidelines for the management and follow-up of these patients. Additionally, most of the studies on the efficacy of treatment are in the adult population and do not take into consideration the role of parents in the healing process. This calls for additional RCTs in the pediatric population to evaluate the efficacy of different treatment plans and to better understand the family’s role in the management of this disorder. Functional neurological disorder has an overall poor prognosis; however, children and young adults appear to have a better outcome. Although some factors appear to have a good correlation with better outcomes (shorter duration of symptoms, early diagnosis, younger age), the difference in follow-up rates in the current literature (50–100%) provides a substantial bias both in patients with better outcomes as well as in those with poorer ones. Prospective studies with longer follow-up rates and periods are needed to evaluate long-term outcomes, including recurrence of FND and development of other functional symptoms or psychiatric disorders. Additionally, data from these studies could be used to develop follow-ups based on prognostic factors, functionality scales, and treatment response. 

## 5. Conclusions

FND is a common issue in the pediatric population, with additional needs compared to the adult population. First, the family must understand the diagnosis and support the child in the rehabilitation program. Family counselling may also be needed to ensure good compliance. Most of the findings in the literature on etiology, prognostic factors, and treatment plans come from adult studies, which are lacking in information on the particularities of pediatric-onset FND. As the concept of FND diagnosis and treatment evolves over time, in-depth reviews and prospective studies on the pediatric population will be needed to improve our understanding of the particularities of childhood-onset FND and to better manage the children who have this disorder.

## Data Availability

Not applicable.

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
