# Peer review of "Functional Neurological Disorder–Old Problem New Perspective"

_ijerph, 2023, doi:10.3390/ijerph20021099_

Round 1
Reviewer 1 Report
The authors reported a review on Functional neurological disorders in childern. I have some comments to the authors:
(i) The authors should ask the help of an english natural speaker
(ii) In the introduction, the authors should include this important study published this year covering the phenomenology of functional motor disorders both in children and adults:
Lidstone SC, et al Functional movement disorder gender, age and phenotype study: a systematic review and individual patient meta-analysis of 4905 cases Journal of Neurology, Neurosurgery & Psychiatry 2022;93:609-616.
(iii) “When suspicion of FND is raised, the medical practitioner should keep in mind that FND can coexist with an organic disease and that not all patients with FND have a history of stress factors or psychiatric comorbidities”. In order to provide proper tools to the readers, the authors should also include this paper:
Tinazzi et al. Functional motor disorders associated with other neurological diseases: Beyond the boundaries of "organic" neurology. European Journal of Neurology. 2021
(iv) Specify that this is a narrative review.
(v) Methods are too short. The authors should provide the PRISMA algorithm, and they should specify inclusion and exclusion criteria for this review.
(vi) In the sub-paragraph related to trauma, in the last part, the authors correctly covered the COVID-19 part. However, the authors should also state that FND was observed after COVID-19 vaccination. I suggest including the first three report of this phenomenon:
1) Butler et al. Functional Neurological Disorder After SARS-CoV-2 Vaccines: Two Case Reports and Discussion of Potential Public Health Implications. 2021
2) Ercoli T et al. Functional neurological disorder after COVID-19 vaccination. 2021
3) Fasano A et al. Functional disorders after COVID-19 vaccine fuel vaccination hesitancy. 2021
(vii) “When possible, the diagnosis should be given to the patients in the presence of a mental health specialist.” I do not agree with this point. The diagnosis should be made by a neurologist or a psychiatrist.
(viii) “However, there is a 4% rate of false-positive diagnosis,”. What is the reference of this affirmation?
Author Response
Thank you for reviewing this manuscript and providing the opportunity to improve our work.
(i) The authors should ask the help of an english natural speaker
(i)The manuscript will be revised by a natural English speaker. Supplementary revisions were made.
(ii) In the introduction, the authors should include this important study published this year covering the phenomenology of functional motor disorders both in children and adults:
Lidstone SC, et al Functional movement disorder gender, age and phenotype study: a systematic review and individual patient meta-analysis of 4905 cases Journal of Neurology, Neurosurgery & Psychiatry 2022;93:609-616.
(ii) We included the provided articles regarding the phenomenology of functional motor disorders in our review.
(iii) “When suspicion of FND is raised, the medical practitioner should keep in mind that FND can coexist with an organic disease and that not all patients with FND have a history of stress factors or psychiatric comorbidities”. In order to provide proper tools to the readers, the authors should also include this paper:
Tinazzi et al. Functional motor disorders associated with other neurological diseases: Beyond the boundaries of "organic" neurology. European Journal of Neurology. 2021
(iii)The references were added.
(iv) Specify that this is a narrative review.
(iv) We completed the methods section to specify that the present manuscript is a narrative review.
(v) Methods are too short. The authors should provide the PRISMA algorithm, and they should specify inclusion and exclusion criteria for this review.
(v) We expanded the methods section. As the article is a narrative review and not a systematic review or meta-analysis, we did not provide PRISMA algorithm.
(vi) In the sub-paragraph related to trauma, in the last part, the authors correctly covered the COVID-19 part. However, the authors should also state that FND was observed after COVID-19 vaccination. I suggest including the first three report of this phenomenon:
1) Butler et al. Functional Neurological Disorder After SARS-CoV-2 Vaccines: Two Case Reports and Discussion of Potential Public Health Implications. 2021
2) Ercoli T et al. Functional neurological disorder after COVID-19 vaccination. 2021
3) Fasano A et al. Functional disorders after COVID-19 vaccine fuel vaccination hesitancy. 2021
(vi) We added a sub-paragraph regarding FND after COVID-19 vaccinations
(vii) “When possible, the diagnosis should be given to the patients in the presence of a mental health specialist.” I do not agree with this point. The diagnosis should be made by a neurologist or a psychiatrist.
(vii) We agree that the diagnosis should be made by a neurologist or a psychiatrist. In our opinion it is helpful to deliver the diagnosis in a multidisciplinary team, including a mental health specialist in addition to the neurologist or psychiatrist to ensure a higher rate of acceptance.
(viii) “However, there is a 4% rate of false-positive diagnosis,”. What is the reference of this affirmation?
(viii) The reference of this affirmation is cited in our manuscript. Reference number 57 in the revised manuscript.
Stone et al., “Systematic review of misdiagnosis of conversion symptoms and ‘hysteria,’” BMJ, vol. 331, no. 7523, p. 989, Oct. 2005, doi: 10.1136/BMJ.38628.466898.55
Reviewer 2 Report
Authors provided a manuscript for the review article titled ” FUNCTIONAL NEUROLOGIC DISORDER – OLD PROBLEM NEW PERSPECTIVE”. The aim of the paper was to provide an update on pediatric FND. Authors conducted a literature search of PubMed and SCOPUS database and reviewed a total of 80 articles. They gathered information on etiology, diagnosis, treatment and prognosis in FNS.
The article is written in a clear language, it is comprehensive and the construction of the work is correct. The discussion is exhaustive and broad. The subject of the paper is interesting and relevant. It underlines the important problem of FNS in paediatric population.
The article presents, however, some limitations. Most of them are naturally associated with the type of the paper which is a review article. Concerns may be associated with potential bias due to literature searching practices that are not presiced, the possibility of the omission of relevant research and the influence of authors' personal viewpoints. Also in this case there are no information provided on the type of studies or articles included in the search. It would be appreciated if authors present any key for choosing the publications used as references for the article.
Another issue is that references are moderately recent. Most of them are dated before 2017. My question for the authors would be whether the search was restricted to any time period and to give the rationale for using so many references that are not up to date.
My other advice to make the article more comprehensive would be to expand the information on clinical presentations of FNS in paediatric patients.
Minor interpunction and spacing errors can be noticed. Proof reading of the manuscript would be advised.
Author Response
Thank you for reviewing this manuscript and providing the opportunity to improve our work.
We expanded the methods section to provide better information on the types of studies and articles included in our search. There was no period for the selection of the articles. The PubMed and SCOPUS data base were searched form the beginning until September 2022. The authors chose the most relevant articles for our review, regardless the date of publication.
A better description of the clinical presentation of FND in paediatric patients was added.
Round 2
Reviewer 2 Report
I believe that authors adequately addressed all the remarks given in previous review. However, I find formatting of the text after the corrections extremely hard to read and confusing. It would be appreciated, if the authors provide the manuscript in a more "clear", edited format.
Author Response
Thank you for reviewing this manuscript and providing the opportunity to improve our work.
We edited the format of the text for a clearer version.